



# Quantifying and reducing researcher subjectivity in the generation of climate indices from documentary sources

5 George C.D. Adamson[1], David J. Nash[2,3], Stefan W. Grab[3]

1 Department of Geography, King's College London, London, United Kingdom
2 School of Applied Sciences, University of Brighton, Brighton, United Kingdom
School of Geography, Archaeology and Environmental Studies, University of the Witwatersrand, Johannesburg, South Africa

*Correspondence to*: George C.D. Adamson (george.adamson@kcl.ac.uk)

**Abstract.** The generation of index-based series of meteorological phenomena, derived from narrative descriptions of weather and climate in historical documentary sources, is a common method to reconstruct past climatic variability. This study is the first to explicitly examine the degree of inter-rater variability in producing such series, a potential source of bias in index-based analyses. Two teams of raters were asked to produce a five-category annual rainfall index series for the same

dataset, consisting of transcribed narrative descriptions of meteorological variability for 11 'rain-years' in nineteenth-century Lesotho, originally collected by Nash and Grab (2010). One group of raters (n = 71) comprised of students studying for postgraduate qualifications in climatology or a related discipline; the second group (n = 6) consisted of professional meteorologists and historical climatologists working in southern Africa. Inter-rater reliability was high for both groups, at r = 0.99 for the student raters and r = 0.94 for the professional raters, although ratings provided by the student group

disproportionately averaged to the central value (0: normal/seasonal rains) where variability was high. Back-calculation of intraclass correlation using the Spearman-Brown prediction formula showed that a target reliability of 0.9 could be obtained with as few as eight student raters, and four professional raters. This number reduced to two when examining a subset of the professional group (n = 4) who had previously published historical climatology papers on southern Africa. We therefore conclude that variability between researchers should be considered minimal where index-based climate reconstructions are

generated by trained historical climatologists working in groups of two or more.

## 1 Introduction

The generation of ordinal-scale indices from documentary sources is one of the most widely used approaches in historical climatology to transform raw weather descriptions into semi-quantitative data (Brázdil et al., 2010; Pfister et al., 2018). The index approach has been applied globally (apart from Antarctica) to reconstruct a variety of meteorological phenomena (Nash



et al., 2021). Temperature is the most commonly analysed parameter, particularly for the northern hemisphere, with multi-centennial index-based series available for various parts of Europe (e.g. Pfister, 1984; Alexandre, 1987; Pfister, 1992; Brázdil and Kotyza, 1995; Ogilvie and Farmer, 1997; Dobrovolný et al., 2009; Glaser and Riemann, 2009; Camuffo et al., 2010), China (e.g. Wang et al., 2001) and the Americas (e.g. Baron et al., 1984; Baron, 1995). Precipitation is the second most widely analysed variable, with notable regional series produced for parts of Europe (e.g. Van Engelen et al., 2001; Pfister et al., 2006;

Rodrigo and Barriendos, 2008; Dobrovolný et al., 2015; Fernández-Fernández et al., 2015; Bauch et al., 2020), China (e.g. Ge et al., 2018), Africa (e.g. Nicholson et al., 2012a; Norrgård, 2015; Nash et al., 2016; Nicholson et al., 2018) and Australia (Fenby and Gergis, 2013; Gergis and Ashcroft, 2013). Index-based series have also been generated for floods (e.g. Pfister, 1999; Glaser and Stangl, 2004; Prieto and Rojas, 2015; Salvisberg, 2017; Kiss, 2019), droughts (e.g. Brázdil et al., 2018; Erfurt and Glaser, 2019), snowfalls (e.g. Ge et al., 2003), storminess (e.g. Wang, 1980; Dominguez-Castro et al., 2019), dust fall (e.g.

Fei et al., 2005) and sea-ice cover (Ogilvie and Jónsson, 2001).

There is no consistent global approach to climate index generation (Nash et al., 2021), the closest being the method developed by Christian Pfister in the 1980s for the creation of Swiss temperature and precipitation series (Pfister, 1984); a method widely adopted across Europe and, in the case of rainfall reconstructions, across Africa (e.g. Vogel, 1989; Nash and Endfield, 2002),

Australia (Fenby and Gergis, 2013) and India (Adamson and Nash, 2014). Under the Pfister method, indices are normally generated at a monthly level through the analysis of contemporary reports of climate and related conditions and, where available, biophysical proxies in historical records. Each month is placed into one of 7 classes ranging from -3 (very cold/dry) to +3 (very warm/wet), which can then be added together to produce seasonal or annual series. Central to the Pfister method – and, indeed, almost all index-based analyses of meteorological variables – is the identification of regionally relevant

phenomena (e.g. the timing and duration of snowfall, or various plant-phenological indicators) that can be used to allocate a specific month to a specific index category. So, for example, a relatively cold March in Switzerland would be one where historical observers described prolonged snow cover and frequent snowfalls, whereas a warm March would be characterised by no snowfall and the early flowering of sweet cherry trees (Pfister, 1992; Pfister et al., 2018). This approach has been adapted for regions with less rich documentary evidence, or a seasonal skew to the available climate descriptions, through a reduction

in the number of index categories (e.g. to five or three classes) and/or the temporal resolution of the reconstruction (to seasonal or annual).

Index-based analyses contain two stages where potential biases can be introduced. First, where plant-phenological indicators are not available in historical sources, the classification of past meteorological conditions becomes entirely reliant on narrative

descriptions of weather and related phenomena. Contemporary weather descriptions can be considered as accurate representations of other places, on the basis that the writers were recording eye-witness accounts and had first hand experiences (Duncan, 1997). However, all narrative accounts necessarily reflect the positionality and personal motivation of the observer, as well as his/her intended audience (Brázdil et al., 2010). The spatiality of representation of 'fact' also needs to be considered



– individual observers may view weather and related environmental conditions for a specific place and time very differently,
while an author's perception of his/her environment may change with increased familiarity and experience (Nash and Endfield, 2008). There can, as a result, be very different descriptions of the same weather 'event' recorded by different observers or the same observer but at different times (Duncan and Gregory, 1999). A further factor of particular concern when weather descriptions are made by 'non-local' authors, relates to the nationality and background of correspondents. Unusual or extreme events are judged by individuals against a 'normal' range of climate variations, itself a function of the lived experience of the individual and the extent of climate variability communicated through oral histories or historical knowledge (cf. Hassan, 2000). Finally, the "data filter of human recollection" may come into play (Bryson and Padoch, 1980, p.585). Subjective references such as 'this has been the wettest summer in memory' can only be considered in relation to the perceived normal, and might record short-term, but probably not long-term, change. Many of these issues can be overcome by a source-critical approach to historical records and careful triangulation between observers, but not all (Brázdil et al., 2005).

The second area of potential bias in index-based analyses comes at the classification stage of index development – this is the primary focus of our study. Here, the accuracy of any index series depends upon the ability of the analyst to interpret and evaluate collections of raw weather descriptions and then categorise them into appropriate classes. As Nash et al. (2021) note, transforming information from historical documents to numbers on a scale requires considerable expertise to minimise subjectivity. Consequently, this should ideally be undertaken by researchers with a good scientific knowledge of the regional climate, and an understanding of the language of the time period during which the sources were written. Nash et al. (2016) argue that, if resources allow, index series should ideally be developed by combining the independent classifications of more than one analyst, with qualitative confidence ratings given for each month, season or year of the series, as appropriate (Kelso and Vogel, 2007). However, to date, the degree of subjectivity produced by a single analyst versus a group has never been explicitly tested.

This study aims to explore and quantify the degree of error between researchers assigning indices to the same historical documentary dataset, and provide recommendations to reduce future bias in index-based series derived from documentary sources. The paper utilises a documentary dataset produced for Lesotho by Nash and Grab (2010), derived primarily from missionary and colonial government descriptions of climate and climate-dependent phenomena during the late nineteenth century. Transcribed descriptive data relating to eleven 'rain-years' (see section 2) were given to two teams of researchers (hereafter referred to as 'raters'): 71 postgraduate students ('the student rater group') and six professional meteorologists or historical climatologists ('the professional rater group'). Each rater was asked to generate rainfall indices for each rain-year on a 5-point classification scale. The paper calculates intraclass correlations (ICC) for each group to determine (i) the degree of inter-rater reliability within the whole group and (ii) the number of raters required to achieve a target reliability. The paper discusses differences in ratings and interclass correlations between the professional and student rater groups, and provides suggestions for minimising researcher subjectivity in future index-derived reconstructions.



## 2 Materials and methods

This study uses a dataset obtained from documentary-based narrative descriptions of rainfall variability collected for Lesotho

(Nash and Grab, 2010). Lesotho is a mountainous country, landlocked within South Africa and situated within the Drakensberg range. Average annual rainfall varies from 735 mm p.a. in the lowlands to 1,600 mm p.a. in the Drakensberg. Approximately 80% of rainfall occurs during the peak austral summer rainy season (December to February), associated with the convergence of easterly and westerly air masses bringing rainfall from the Indian and Atlantic Oceans (Tyson, 1986; Sene et al., 1998). The winter months are dominated by high pressure over the central interior, with minimum rainfall from May to August. As the

peak rainy season straddles more than one calendar year, Nash and Grab (2010) grouped the narrative descriptions according to 'rain-year' (encompassing the period July to June) to generate their rainfall reconstruction, rather than calendar year. Nash and Grab (2010) analysed materials from 76 rain-years (1824-1900); of these 11 are used for our current analysis. The documentary materials analysed for the original study included letters, reports and journals held in the archives of missionary societies in London, Oxford and Maseru, newspapers held at archives in Morija, and government records held in various

collections. Passages that alluded to weather/climate conditions, either directly or indirectly (for example state of river flow, crops etc), were recorded verbatim, with French- and Sesotho-language materials translated into English. See Nash and Grab (2010) for full details of sources and data collection methods.

Raters were given verbatim English-language transcriptions (or translations) of all references to climate variability and climate-

related phenomena recorded for the rain-years 1889/90 to 1899/1900 inclusive, as per Nash and Grab (2010). Raters were also provided with the following contextual materials:

- An introduction to the exercise, including a map showing the location of Lesotho within southern Africa.
- Pages 618 to 623 of Nash and Grab (2010), comprising an introduction to the study, a map of Lesotho with locations from which evidence was obtained, an explanation of the regional climatology, details of documentary sources

(including how they were collected and where from), and an explanation of the methodology used.
- The section 'Methods of Analysis' from pages 625-626 of Nash and Endfield (2002), which includes further detail on the reconstruction methodology.

Following the methodology detailed in Nash and Endfield (2002) and Nash and Grab (2010), raters were asked to categorise

each rain-year into one of five ordinal classes based on their assessment of the documentary materials: very wet/floods (+2), relatively wet (+1), normal (seasonal rains) (0), relatively dry (-1) and very dry/drought (-2). Raters were also asked to provide a brief explanation of why they assigned each rain-year to a particular category, and a summary of rainfall conditions. This approach to index development represents a variation on the 'Pfister method' (see Pfister et al., 2018; Nash et al., 2021), using five categories rather than seven, and generating a series for an entire season rather than monthly. It was first developed by

Vogel (1989) as a method for reconstructing rainfall in the Eastern and Southern Cape regions of South Africa, and has been widely used for the derivation of rainfall indices in southern Africa and monsoon regions of India.

Raters were divided into two groups: 71 postgraduate students, and six professional scientists. The student rater group consisted of postgraduate students based in London, studying for a masters or doctoral qualification in a climate or environmental
discipline. All raters had first degrees in related disciplines, and some experience of conducting scientific research. 56 student raters were studying for masters qualifications at King's College London and registered either on the modules 'Global Environmental Change: Past and Present' or 'Climate: Science and History'; these students undertook the rainfall classification individually. The other 15 raters were first year postgraduate research students registered on the London NERC Doctoral Training Partnership. Students in this group undertook the activity in groups of two, although each pair is counted as an
individual rater for the purposes of this study (that is, 15 'raters', but 30 students in total). The professional group consisted of four published historical climatologists who had derived similar rainfall series in other parts of southern Africa (Kelso and Vogel, 2007; Neukom et al., 2014; Hannaford and Nash, 2016; Nash et al., 2016; Grab and Zumthurm, 2018; Nash et al., 2018), and two senior meteorologists at the South African and Lesotho Meteorological Services, both of whom were former MSc students studying climatology at the University of the Witwatersrand.


**3 Results**

**3.1 Student Rater Group**

**3.1.2 Summary Statistics**

Summary statistics for the student rater group are provided in Table 1. The three statistics of central tendency (mean, median
and mode) give very similar results, with only 1896-97 showing differences, and only by one category (median 0, mode -1, mean 0). We consider the mode to be the most robust statistic for this exercise, due to the tendency for both median and mean to reduce ranks to the centre (that is, 0: normal rainfall) where there is large variation between raters. The mean is likely to be unsuitable given the ordinal nature of the data, but is included here for comparison. Of the 11 rain-years, the closest agreement between raters was for 1890/91, for which only 10 of the 71 raters assigned a category other than +2, and these raters all
assigned a rating of +1. In this case, most raters justified their rating on the basis of considerable reporting on widespread flooding and 'abnormal' rainfall. The rain-year 1894/95 also showed close levels of agreement, with 63 raters assigning a category of 0. Here, raters ordinarily justified their decision based on reports of a plentiful harvest and a general lack of specific references to rainfall.





**Table 1:** Summary statistics of index values applied to the eleven rain-years by the 71 postgraduate student raters. Index values in Nash and Grab (2010) are provided for comparison. Mean and median values are rounded to the nearest integer.

| Rain-year | Median | Mode | Sample mean | Standard deviation | *Nash and Grab (2010)* |
|---|---|---|---|---|---|
| 1889/90 | 0 | 0 | 0 | 0.89 | *-1* |
| 1890/91 | +2 | +2 | +2 | 0.35 | *+2* |
| 1891/92 | 0 | 0 | 0 | 0.83 | *+1* |
| 1892/93 | 0 | 0 | 0 | 0.96 | *+1* |
| 1893/94 | +1 | +1 | +1 | 0.64 | *+1* |
| 1894/95 | 0 | 0 | 0 | 0.44 | *0* |
| 1895/96 | -1 | -1 | -1 | 0.8 | *-1* |
| 1896/97 | 0 | -1 | 0 | 0.95 | *-1* |
| 1897/98 | -1 | -1 | -1 | 1.16 | *-1* |
| 1898/99 | -1 | -1 | -1 | 0.89 | *-1* |
| 1899/1900 | 0 | 0 | 0 | 0.89 | *0* |

Rain-years showing the greatest degree of rater variability were 1897/98 (stdev = 1.16), 1892/93 (0.96), 1896/97 (0.95), 1899/1900 (0.89), 1889/90 (0.89), and 1891/92 (0.83). In each of these cases, raters selected the full range of rankings, from -2 to +2. In nearly all these years, the documentary materials mentioned either spatial variability (with drought in some places and heavy rain in others) or temporal variability (with late but heavy rains). The exception was 1891/92, for which raters were primarily confused by references to a plague of locusts, which was interpreted either as reflecting drought (evidenced by poor crops), or as evidence for heavy rainfall (producing vegetation/good crops suitable for locusts). Notably, all these years, with the exception of 1897/98, produced medians that reduced to the centre. This was not the case with Nash and Grab (2010), who had rated these years either as +1 or -1.

### 3.1.2 Inter-Rater Reliability

Inter-rater reliability was calculated using the Two-Way Random Intraclass Correlation Coefficient (ICC) (Landers, 2015) using the IBM SPSS software. ICC provides for a measure of the degree of random variation in scoring, and the proportion of ratings that represent an underlying construct. Following Portney and Watkins (2007), ICC values of ≥ 0.5 are taken to represent moderate reliability, values ≥ 0.75 to represent good reliability, and values ≥ 0.9 to represent excellent reliability.

The overall ICC for all student raters (n = 71) was 0.99, considered to represent excellent inter-rater reliability. This implies a high degree of internal consistency between raters; however, it is important to note that this only measures reliability between





raters, and does not give an indication of the relationship between the recorded index value and the 'real' rainfall, or rainfall as recorded by a rain gauge or some other accurate measuring device.


As n = 71 is a very high number of raters, the Spearman-Brown prediction formula was used to determine the minimum number of raters within the sample required to produce a target level of reliability. To calculate $n$ reliability the formula is as follows, where $n$ = number of raters, $\rho_{xx'}^*$ is predicted reliability, and $\rho_{xx'}$ is the reliability for k (71) raters:


$$n = \frac{\rho_{xx'}^*(1 - \rho_{xx'})}{\rho_{xx'}(1 - \rho_{xx'}^*)}$$

Results are presented in Table 2. 'Moderate' reliability ($\rho \geq 0.5$) can be achieved by selecting one rater at random from the sample of 71; the calculated IRR for one rater within the sample was 0.54. 'Good' inter-rater reliability ($\rho \geq 0.75$) is achieved with three or more raters, and 'excellent' reliability with eight or more. This suggests that very large numbers of raters are not

required to provide high levels of internal consistency in the generation of climate indices. Variation between raters can be minimised by selecting eight raters or more, even where the raters are non-specialists.

**Table 2:** Number of raters within the larger sample of student raters required to achieve a target ICC value. Calculated from the ICC for the entire sample of student raters (n = 71) using the Spearman-Brown prediction formula. Number of raters are
rounded up to the nearest integer.

| Target reliability | 0.5 | 0.75 | 0.8 | 0.85 | 0.9 | 0.95 | 0.99 |
|---|---|---|---|---|---|---|---|
| Minimum number of raters | 1 | 3 | 4 | 5 | 8 | 17 | 85 |

### 3.2 Professional Rater Group

### 3.2.1 Summary Statistics

Whilst the inter-rater reliability for the student rater group was very high, it was notable that six of the eleven rain-years
produced values of zero, compared to two years in Nash and Grab (2010). This may represent a tendency for average rankings to reduce to the centre due to variability between raters. It is well-documented within the historical climatology literature that the generation of climate index values requires careful interpretation by trained researchers (see, for example, Ingram et al., 1981; Glaser, 1996; Nash et al., 2018; Pfister et al., 2018). Although the student sample described in section 3.1 included individuals that were all conducting higher studies or research in climate science or a related discipline, they were not
specialists in historical climatology. Nonspecialist raters may misinterpret documentary materials; for example, failing to take into account the familiarity of the original writer with the area, or potential biases that may encourage them to over- or under-





report rainfall intensity. These are particular issues with this dataset, where writers were primarily European missionaries who, at least at first, may have described conditions based on their past Euro-centric climate experiences. Additionally, whilst the students were provided with an introduction to the climate of Lesotho through Nash and Grab (2010), most were otherwise
unfamiliar with the region.

The exercise was therefore repeated with six professional scientists, four of whom were historical climatologists who had published research with a focus on southern Africa, and two of whom were senior forecasters in meteorological services in South Africa and Lesotho. Summary statistics for these researchers are presented in Table 3. Despite the small sample size, 7
of the 11 years have standard deviations lower than the student rater group, indicative of greater agreement amongst the professional raters. Ratings are more similar to those given by Nash and Grab (2010), with only 1891/92 producing a different rating (0, versus +1 by Nash and Grab 2010, although one of the six researchers also gave the year a +1 rating).

**Table 3:** Summary statistics of index values applied to the eleven rain-years by the 6 professional raters. Nash and Grab (2010)
and the median/mode of the student raters (from Table 1) are provided for comparison.

| Rain-year | Median | Mode | Sample mean | Standard deviation | *Nash and Grab (2010)* | *Median: student raters* | *Mode: student raters* |
|-----------|--------|------|-------------|--------------------|------------------------|--------------------------|------------------------|
| 1889/90 | -1 | -1 | -1 | 0.84 | *-1* | *0* | *0* |
| 1890/91 | +2 | +2 | +2 | 0 | *+2* | *+2* | *+2* |
| 1891/92 | 0 | 0 | 0 | 0.98 | *+1* | *0* | *0* |
| 1892/93 | +1 | +1 | +1 | 0.52 | *+1* | *0* | *0* |
| 1893/94 | +1 | +1 | +1 | 0.89 | *+1* | *+1* | *+1* |
| 1894/95 | 0 | 0 | 0 | 0.52 | *0* | *0* | *0* |
| 1895/96 | -1 | -1 | -1 | 0.75 | *-1* | *-1* | *-1* |
| 1896/97 | -1 | -1 | -1 | 0.41 | *-1* | *0* | *-1* |
| 1897/98 | -1 | -1 | -1 | 0.55 | *-1* | *-1* | *-1* |
| 1898/99 | -1 | -1 | -1 | 0.55 | *-1* | *-1* | *-1* |
| 1899/1900 | 0 | 0 | 0 | 1.05 | *0* | *0* | *0* |

Median and mode values differ between the professional group and student group for several years. For 1889/90, the professional group produced a value of -1 instead of 0, for 1892/93 +1 instead of 0, and for 1896/97 -1 instead of 0 (although the mode of the student raters for this year was also -1). Each of these years showed relatively high standard deviations in the
results from the student raters and each had provided median values of 0, suggesting that the student raters produced a wide





spread of results that had reduced to zero. For each of these years the professional group provided values equal to the Nash and Grab (2010) series.

### 3.2.2 Inter-Rater Reliability

The overall ICC for the professional raters (n = 6) was 0.94. When assessing the professional historical climatologists (n = 4)
as a single group, this rose to 0.95. This suggests that training in historical climatology methods may be particularly important, as well as detailed knowledge of the climatology of the region in question. The calculated reliability for a single rater was 0.73 for the professional group, and 0.83 for the historical climatologist sub-group. The minimum number of raters required to achieve a target level of reliability for the whole group of professional raters and for the subsample of historical climatologists is provided in Table 4. The results suggest that high internal consistency can be achieved for a group of 4 raters who are highly
trained in the climatology of the region. If these raters are also trained as historical climatologists, high reliability can be achieved from an average of two researchers generating separate reconstructions. In practice, this is already common in historical climatology, where researchers often work in teams of two or more.

**Table 4:** Number of raters within the sample of professional raters required to achieve a target value ICC value. Values are
provided for all professional raters (n = 6) and for the subsample of trained historical climatologists (n = 4).

| Target reliability | 0.5 | 0.75 | 0.8 | 0.85 | 0.9 | 0.95 | 0.99 |
|---|---|---|---|---|---|---|---|
| **Minimum number of raters – professional raters group** | 1 | 2 | 2 | 3 | 4 | 8 | 37 |
| **Minimum number of raters – historical climatologists subgroup** | 1 | 1 | 1 | 2 | 2 | 4 | 20 |

### 4 Discussion

We have shown that the issue of researcher subjectivity in the generation of climate index series from documentary sources can be minimised by repeating the process of index generation with a small team of researchers and taking a measure of central
tendency. We would advise using the mode, as this is least likely to reduce to the centre (i.e., produce values of 0) in years where there is considerable disagreement between researchers. However, the index values generated for this project using the median had only minor differences to those produced from the mode, and mean values will be suitable when only two researchers have generated indexes. Our results suggest that high levels of inter-rater reliability can be achieved with teams of two researchers or more, if the researchers are trained in historical climatology methods and have detailed knowledge of the
climate of the region. The results are consistent with the findings of Ge et al. (2008) on documentary-derived temperature series in China. These authors found that temperature series tended to correlate and cluster most closely where they related to



regions with a close geographical proximity, suggesting that regional climatic differences are more important than differences in interpretation in explaining discrepancy between series.

In general, years that produced the greatest variability of results within the student rater group were those where the rainfall characteristics at the start and end of the season differed. For example, the historical climatologists assigned a rating of -1 (relatively dry) to 1889/90, drawing particular significance to overall reports of drought and the lateness of rain. Student raters gave mixed ratings, with some raters giving more weight to reports of drought, and others to reports of heavy rains towards the end of the season. Conversely in 1892/93, the historical climatologist group put considerable weight on the reported

intensity of late rains to assign a category of +1. During this year many of the student groups gave negative (-2/-1) or average (0) ratings, due reports of poor crops and famine during the early part of the season. Historical climatologists assigned a classification of -1 to 1896/97, in part because of reports of drought in the early part of the following rain-year, whereas the student groups showed little evidence that they had considered cross-season indicators of rainfall variability. Each of these years produced a rating of 0 from the student group, suggesting a high degree of variability and a reduction to the central value.

Conversely, the professional group produced values of +1/-1, as did Nash and Grab (2010).

One area that seems to have caused particular variation within the student group, was the interpretation of diary entries written by Sir Godfrey Lagden, the colonial Resident Commissioner for Basotholand, between 1884 and 1900. The majority of these entries consisted of one-line accounts of daily weather conditions or the impacts of weather events on daily activities. The

historical climatologists largely did not mention the diaries, but they seemingly had much influence on many student raters; some student raters noted, for example, the frequency of storms or applied a high weighting to Lagden's specific terminology to describe rainfall (e.g. 'slight rain' or 'little rain'). This confirms the importance of training in the interpretation and analysis of historical sources for the derivation of indices, also demonstrated by the substantially higher inter-rater reliability within the historical climatologist group compared to the student group. Greater variability within the student group was also likely

affected by the unfamiliarity of many with the climate of the region, beyond the background information from Nash and Grab (2010). Data from the professional group, however, suggests that training in historical climatology is important as well as knowledge of local climatology or meteorology. All members of this group were trained climatologists and/or meteorologists (three were resident in the area), of which one had received specific training as a historical climatologist and had published in this subdiscipline. By far the strongest agreement was between the four historical climatologists, with ICC increasing when

viewing these four raters as a stand-alone group.

Overall, this study demonstrates that inter-rater reliability is highest when index series are generated by researchers trained in historical climatology and with a working knowledge of the climate of the region. The calculated reliability of a single rater within the historical climatologist group (n = 4) was 0.83, rising to 0.90 for two raters and 0.94 for three raters. This should

provide additional confidence to previous index series produced in small teams, as is common in historical climatology. In



future, echoing Nash et al. (2016), we suggest that researchers working on climate index development in small teams should produce individual reconstructions, with the final published index values generated by a measure of central tendency. For complete transparency, it would also help if each individual reconstruction is presented in the published results.

It should be noted that the results presented here give no indication of the reliability of the index values in relation to 'real' rainfall, or rainfall measured by accurate rain gauges. Indices derived from documentary data are reliant on the materials available, which may be incomplete, fragmentary, often not relating directly to climate and subject to the biases of the observer. Statistical methods have been developed to calibrate these records, generally using a crossover instrumental record (see Brázdil et al., 2010; Dobrovolný, 2018, for a summary). Calibration is, however, highly sensitive to variability within the

reconstruction process itself, which our results suggest can be reduced substantially by using two or more raters. We note that such instrumental rainfall series are available for Lesotho (Maseru) and the region south of Lesotho (Aliwal North) for the time period in question. However, the series are incomplete and do not covary, so would not permit such a process for these records.

Care should also be taken in interpreting our results beyond this particular dataset without replication of the study. Repeating the exercise with other sets of documentary records would be helpful, particularly those from mid- and high-latitude countries that do not have a single rain season. Climate index series in these regions are commonly generated by month rather than season, with indices summed or averaged to give an annual value (see Pfister et al., 2018), providing both more scope for error and more likelihood that variability will be reduced in the averaging of multiple index values for a year. It would therefore be

beneficial to repeat the exercise following the standard Pfister method, which also commonly ranks against a 7-point scale rather than a 5-point scale.

**5 Conclusions**

This study set out to explore and quantify the degree of error between individuals generating ordinal-scale climate indices from a historical documentary dataset. Ultimately, our findings reinforce the conclusions of Nash et al. (2021) that climate indices

should be derived by those with training in historical climatology methods, as well as a detailed knowledge of both the climate of the area in question and the nature of the documentary sources. However, we have demonstrated that generating such indices using groups of two or more researchers will substantially reduce the issue of researcher subjectivity. Our results also suggest that researchers who are experienced historical climatologists are likely to produce similar indices from the same dataset, which has never been explicitly demonstrated previously.


Our results have implications for the utilisation of citizen science approaches to generate climate indices from documentary data. Subjectivity can be reduced by taking a measure of central tendency (ideally the mode) from small groups of citizen scientists. Our results suggest that 'good' inter-rater reliability can be achieved with three raters, and 'excellent' with eight





raters, although it should be reiterated that our raters had at least a strong first degree in an environmental discipline. However,

researchers seeking to utilise citizen science should be cogent of the tendency for ratings to reduce to the central value (in this

case 0: normal) in years where there is a high degree of variability between raters, particularly where rainfall is not consistent

throughout a season. Overall, the results reiterate the importance of training in historical climatology, and suggest that

confidence should be given to previous reconstructions produced by small teams of historical climatologists.

**Data Availability**

The full dataset of transcribed nineteenth-century documentary references to weather and climate collected for Lesotho by

Nash and Grab (2010) are available on request.

**Author Contribution**

The experiments were conducted and organised by Adamson, using a dataset collected by Nash and Grab. All authors compiled

the manuscript.

**Competing Interests**

The authors declare no competing interests.


**Acknowledgements**

The authors would like to thank all of the students who participated in the study, consisting of postgraduate students at King's

College London on the MSc Global Environmental Change, London NERC DTP, and on the module Climate: Science and

History. We would also like to thank the following professional scientists for their generous participation: Matthew Hannaford,

Clare Kelso, Puseletso Mofokeng, Retsipile Neko, Tizian Zumthurm. The original dataset for this study was collected through

British Academy Small Grant SG-40838.

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
