# Peer review of "Quantifying and reducing researcher subjectivity in the generation of climate indices from documentary sources"

_Climate of the Past, 2021_

## Community Comment (CC1)

**Review of Adamson et al: Quantifying and reducing researcher subjectivity in the generation of climate indices from documentary sources**

The study aims exploring and quantifying the degree of error between researchers assigning ordinal-scale indices to a historical documentary dataset. Two teams of raters were asked to produce a five-category annual rainfall index series for a dataset consisting of transcribed narrative descriptions of meteorological variability for 11 rain-years' in nineteenth-century Lesotho. The authors conclude that variability between researchers should be considered minimal where index-based climate reconstructions are generated by trained historical climatologists working in groups of two or more.

**The study should be accepted with small changes indicated below**:

1. The different preconditions in the derivation of temperature and precipitation indices should be worked out more clearly :

Lines 49-56
 "This approach has been adapted for regions with less rich documentary evidence", **"The Pfister is approach is mainly tailored to reconstructing temperatures for regions with rich documentary evidence and long series of homogenized instrumental measurements (e.g. Pfister, Wanner 2021). In such cases proxy information often allows estimating temperatures for specific months or seasons by using the calibration verification approach (e.g. Dobrovolný 2010). In such cases, the potential bias in classification is very small for trained historical climatologists, as the narrative record and the proxy need to be consistent and meteorologically meaningful.** This approach has been adapted for regions with less rich documentary evidence, or a seasonal skew to the available climate descriptions, through a reduction in the number of index categories (e.g. to five or three classes) and/or the temporal resolution of the reconstruction (to seasonal or annual).
**The situation is different for classifying precipitation. Proxy-data such as information on floods and droughts or the number of rain-days may hardly be calibrated, as precipitation is rather small-scaled in comparison to temperatures and because long homogenized instrumental series of precipitation are quasi non-existent. The study by Dobrovolný et al.(2015), which is perhaps the most sophisticated approach of this kind in Europe, only indicated acceptable reconstruction skill for seasonal precipitation indices in JJA and annual values.**

2. It should be worked out more clearly, also in the abstract, why estimates of historical precipitation conditions in a country of the Global South are significant for the present situation.
3. a map should be included showing the location of Lesotho in southern Africa.

4. an example of a source illustrating the nature of the narratives should be included.

**Suggestion for small changes**:

Line 45: "Under the Pfister method, indices are normally" might be replaced by "**The Pfister Indices, as Mauelshagen (2010) named them**", are normally generated,

Lines 49-51: "relevant phenomena (e.g. the timing and duration of snowfall, or various plant-phenological indicators) "might be replaced by . "regionally relevant **proxy data (e.g. plant-phenological observations, the duration of snow-cover and the freezing of water bodies)**…

Additional references:
Mauelshagen, F. (2010), Klimageschichte der Neuzeit, 1500–1900. Darmstadt: Wissenschaftliche. Buchgesellschaft, 2010
Pfister C. and Wanner H. (2021). Climate and Society in Europe,

---

## Author Response (AR1)

**Response to reviewers**

**#RC1**

The study aims exploring and quantifying the degree of error between researchers assigning ordinal-scale indices to a historical documentary dataset. Two teams of raters were asked to produce a five-category annual rainfall index series for a dataset consisting of transcribed narrative descriptions of meteorological variability for 11 rain-years' in nineteenth-century Lesotho. The authors conclude that variability between researchers should be considered minimal where index-based climate reconstructions are generated by trained historical climatologists working in groups of two or more.

The study should be accepted with small changes indicated below:

The different preconditions in the derivation of temperature and precipitation indices should be worked out more clearly:

Lines 49-56

"This approach has been adapted for regions with less rich documentary evidence", "The Pfister is approach is mainly tailored to reconstructing temperatures for regions with rich documentary evidence and long series of homogenized instrumental measurements (e.g. Pfister, Wanner 2021). In such cases proxy information often allows estimating temperatures for specific months or seasons by using the calibration verification approach (e.g. Dobrovolný 2010). In such cases, the potential bias in classification is very small for trained historical climatologists, as the narrative record and the proxy need to be consistent and meteorologically meaningful. This approach has been adapted for regions with less rich documentary evidence, or a seasonal skew to the available climate descriptions, through a reduction in the number of index categories (e.g. to five or three classes) and/or the temporal resolution of the reconstruction (to seasonal or annual).

The situation is different for classifying precipitation. Proxy-data such as information on floods and droughts or the number of rain-days may hardly be calibrated, as precipitation is rather small-scaled in comparison to temperatures and because long homogenized instrumental series of precipitation are quasi non-existent. The study by Dobrovolný et al.(2015), which is perhaps the most sophisticated approach of this kind in Europe, only indicated acceptable reconstruction skill for seasonal precipitation indices in JJA and annual values.

*Response: Thank you for this observation. This distinction is important but not central to the main aim of the paper. We have amended all references to the 'Pfister Method' to 'Pfister Indices', and added some brief text to the middle and end of para 2 in section 1, as follows:*

*(middle) "The method for generating Pfister Indices is mainly tailored to reconstructing temperature variability for regions with rich documentary evidence and long series of instrumental data (Pfister and Wanner, 2021). Central to the method…"*

*(end) "The reconstruction of Pfister Indices for precipitation is more challenging, since (i) rainfall often varies over smaller spatial scales than temperature, (ii) proxy data such as drought or flood magnitudes are less easy to calibrate, and (iii) the long instrumental series required for calibration are less common than those for temperature. The study by Dobrovolný et al. (2015) of precipitation variability over the last 500 years in central Europe, for example, only identified an acceptable level of reconstruction skill for seasonal precipitation indices in JJA and for annual precipitation values."*

It should be worked out more clearly, also in the abstract, why estimates of historical precipitation conditions in a country of the Global South are significant for the present situation.

*Response: We have added the phrase "…and effectively extend the instrumental record" to the opening sentence of the abstract. We have inserted the following sentence at the end of para 1 in section 1: "The reconstruction of climate indices is a useful tool for examining climate variability during the pre-instrumental period, and is particularly valuable for regions, including many in the Global South, where lengthy meteorological records are lacking."*

A map should be included showing the location of Lesotho in southern Africa.

*Response: We have inserted a map as Figure 1 and added a caption to the manuscript.*

An example of a source illustrating the nature of the narratives should be included.

*Response: We have inserted an image of an example source as Figure 2 and added a caption to the manuscript.*

Suggestion for small changes:

Line 45: "Under the Pfister method, indices are normally" might be replaced by "The Pfister Indices, as Mauelshagen (2010) named them", are normally generated,

*Response: Thanks for this suggestion. This sentence now reads: "Pfister Indices, as named by Mauelshagen (2010), are normally generated…"*

Lines 49-51: "relevant phenomena (e.g. the timing and duration of snowfall, or various plant-phenological indicators) "might be replaced by . "regionally relevant proxy data (e.g. plant-phenological observations, the duration of snow-cover and the freezing of water bodies)…

*Response: This sentence has been amended as suggested.*

Additional references:

Mauelshagen, F. (2010), Klimageschichte der Neuzeit, 1500–1900. Darmstadt: Wissenschaftliche. Buchgesellschaft, 2010

Pfister C. and Wanner H. (2021). Climate and Society in Europe

*Response: These have been added to the manuscript.*

**#RC2**

The article presents interesting results of a remarkable experiment between students and historical climatologists. The possibility of repeating the experiment dealing with a production of annual rainfall index series by two groups of raters with different experience is an indisputable advantage that can bring new findings. The article itself can serve as a good inspiration for other researchers and institutions occupied with historical climatology and having access to daily/monthly weather observations. The future possible experiments can be focused on different variable, area or time scale as the authors stated at the end of the discussion.

From the general point of view, the paper is written comprehensibly and no serious shortages haven´t been noticed. The applied methods seem to be suitable and the results

are clearly presented. Therefore, the study should be accepted and only several suggestions and questions are given for consideration.

*Response: Many thanks for your positive comments.*

Page 6: It would be better if the ICC values were expressed via interval, i.e. "ICC values of 0.5–0.74 are taken to represent moderate reliability, values between 0.75 and 0.89 to represent good reliability, and values ≥ 0.9 to represent excellent reliability."

*Response: We have made this change as suggested.*

Page 7, line 188: "IRR" abbreviation should be explained.

*Response: We have changed this to 'inter-rater reliability' (it was a legacy from a previous draft).*

Why did the authors decide to apply target reliability just of 0.9? Even though the reliability of 0.9 is commonly used in many studies did the author´s decision coincide with ICC values ≥ 0.9 representing excellent reliability? If did this fact should be emphasized at least by one sentence.

*Response: We have clarified this as follows:*

*The results suggest that a target ICC of 0.9 – considered 'excellent' inter-rater reliability by Portney and Watkins (2007) – can be achieved for a group of 4 raters who are highly trained in the climatology of the region.*

I miss a picture of Lesotho in the paper because not every reader can be aware of its precise localization within Africa.

*Response: We have inserted a map as Figure 1.*

I suggest joining a picture of a used documentary source to freshen up the paper.

*Response: We have inserted an image of an example source as Figure 2.*

Did the authors find out what was the average time to process your task by one student/historical climatologist and if time differed significantly? It could be also an interesting point in your study.

*Response: This is a good point. We have added the following sentence to the end of the methodology, and our analysis shows that this had negligible impact on the results:*

*Students were given a two-hour window to complete the reconstruction. The professional group were not time-limited, but only two rater reported spending more than two hours on the analysis, with a median time of 1.5 hours.*

**#CC1**

These comments, posted by Christian Pfister, are identical to those posted under #RC1.